# Explainable mortality prediction models incorporating social health determinants and physical frailty for heart failure patients

Zhenyue Gao[1,2☯], Xiaoli Liu[1,3☯], Yu Kang[4], Pan Hu[5,6], Xiu Zhang[7], Mengwei Li[1], Yumeng Peng[1], Wei Yan[8], Muyang Yan[8], Pengming Yu[7], Zhengbo Zhang[1‡], Qing Zhang[4‡], Wendong Xiao[2]*

1 Medical Innovation Research Department, The General Hospital of PLA, Beijing, China, 2 Beijing Engineering Research Center of Industrial Spectrum Imaging, School of Automation and Electrical Engineering, University of Science and Technology Beijing, Beijing, China, 3 School of Biological Science and Medical Engineering, Beihang University, Beijing, China, 4 Department of Cardiology, West China Hospital, Sichuan University, Chengdu, Sichuan Province, China, 5 Department of Anesthesiology, The 920 Hospital of Joint Logistic Support Force of Chinese PLA, Kunming Yunnan, China, 6 Department of Critical Care Medicine, The First Medical Center, The General Hospital of PLA, Beijing, China, 7 Rehabilitation Medicine Center, West China Hospital, Sichuan University, Chengdu, Sichuan Province, China, 8 Department of Hyperbaric Oxygen Therapy, the First Medical Center, Chinese PLA General Hospital, Beijing, China

☯ These authors contributed equally to this work.
‡ ZZ and QZ also contributed equally to this work.
* wdxiao@ustb.edu.cn

## Abstract

There is limited evidence on how social determinants of health (SDOH) and physical frailty (PF) influence mortality prediction in heart failure (HF), particularly for in-hospital, 90-day, and 1-year outcomes. This study aims to develop explainable machine learning (ML) models to assess the prognostic value of SDOH and PF at multiple time points. We analyzed data from adult patients admitted to the intensive care unit (ICU) for the first time with a diagnosis of HF. Key variables extracted from electronic health records included SDOH (e.g., primary language, insurance type), PF indicators (Braden mobility, nutrition, activity, and fall risk scores), vital signs, laboratory tests, and lung sounds (LS) from both ICU admission and discharge. We employed the eXtreme Gradient Boosting (XGBoost) algorithm to build models for short- and long-term mortality prediction, and used SHapley Additive exPlanations (SHAP) to interpret model outputs and quantify the importance of each feature. The observed mortality rates were 14.8% in-hospital (n = 12,856), 7.0% at 90 days (n = 10,990), and 13.5% at 1 year (n = 10,221). The prediction models achieved area under the receiver operating characteristic curve (AUROC) scores of 0.836 (95% CI: 0.831–0.844) for in-hospital, 0.790 (95% CI: 0.780–0.800) for 90-day, and 0.789 (95% CI: 0.780–0.799) for 1-year mortality. These models outperformed baseline ML algorithms and conventional clinical risk scores. Key predictors of HF outcomes included age, fall risk, primary language, blood urea nitrogen, comorbidities, urine

**Data availability statement:** The clinical dataset used for this work can be fully generated by combining the openly accessible MIMIC-III v1.4 (https://www.physionet.org/content/mimiciii/1.4/) and MIMIC-IV v1.0 (https://physionet.org/content/mimiciv/1.0/) using credentialed access. Source code will be available at https://github.com/gaozhenyue/heart-failure-mortality-prediction.

**Funding:** "This work was supported in part by the National Natural Science Foundations of China (NSFC) under Grant 62171471, Beijing Natural Science Foundation under Grant 7252298 and 7252299, and the Beijing Municipal Science and Technology Project under Grant Z241100007724003." The funders had no role in study design, data collection and analysis, decision to publish, or preparation of the manuscript.

**Competing interests:** The authors have declared that no competing interests exist.

**Abbreviations:** HF, Heart failure; SDOH, Social determinants of health; PF, Physical frailty; ICU, Intensive care unit; LS, Lung sounds; ML, Machine learning; AUROC, Area under the receiver operating characteristic curve; BMI, body mass index; CCI, Charlson Comorbidity Index; SBP, systolic blood pressure; GCS, Glasgow Coma Scale; BUN, blood urea nitrogen; IQR, medians and interquartile ranges.

output, insurance type, and LS findings. Incorporating PF at ICU admission and discharge, along with SDOH such as language proficiency and insurance status, could enhance the identification of high-risk HF patients and may inform targeted interventions.

## Introduction

Heart failure (HF) is a growing global health burden, affecting an estimated 64 million adults worldwide and contributing substantially to hospitalizations and healthcare costs [1–3]. Up to 25–50% of hospitalized HF patients require admission to the intensive care unit (ICU), where they face high risks of in-hospital and post-discharge mortality [4,5]. Despite advances in treatment, HF remains the leading cause of death among cardiovascular diseases globally [6]. Accurate mortality prediction is essential for improving HF management, particularly in identifying high-risk patients and allocating healthcare resources effectively [7,8].

Several risk stratification tools for HF have been developed, including the Get With The Guidelines–Heart Failure (GWTG-HF), ADHERE, and OPTIMIZE-HF scores [9–11]. These models, grounded in traditional linear regression, offer interpretability but are limited in their ability to capture complex, nonlinear interactions among risk factors. In contrast, machine learning (ML) approaches have shown promise in enhancing predictive performance by modeling high-dimensional, nonlinear relationships. However, many ML models overlook key prognostic factors such as social determinants of health (SDOH), physical frailty (PF), and lung sounds (LS), which are increasingly recognized as influential in HF outcomes [12].

SDOH—including socioeconomic status, access to care, and language proficiency—significantly impact disease progression and outcomes in HF [13,14]. Language barriers, in particular, can hinder effective communication during care transitions and discharge, leading to higher readmission rates [15,16]. In the United States, where over 26 million individuals have limited English proficiency, the prognostic role of language remains underexplored in both short- and long-term HF outcomes [15,17]. Insurance type (e.g., Medicare, Medicaid) may also reflect socioeconomic vulnerability, yet its utility as a stratification factor in HF prognosis is not well established [18,19].

PF further complicates HF management. Frailty, characterized by reduced physiological reserve and increased vulnerability to stressors, affects up to 79% of HF patients and is associated with worse clinical outcomes [19,20]. While various tools assess frailty, the Braden Scale—routinely used by nurses—offers a practical measure encompassing mobility, nutrition, and fall risk [21,22]. Previous studies have identified components of the Braden Scale, including fall risk, as independent predictors of mortality in HF [22,23]. Nevertheless, research in this area is limited by small sample sizes, restricted variable sets, and inconsistent outcome definitions.

LS also hold potential prognostic value in HF. Abnormal auscultatory findings such as crackles are common during HF exacerbations and may precede clinical deterioration [24,25]. Despite their relevance, LS are rarely incorporated into existing predictive models.

To address these limitations, there is a critical need for advanced, explainable ML models that integrate SDOH, PF, and LS to enhance risk stratification across care timelines. In this study, we aimed to develop and validate interpretable ML models to predict in-hospital, 90-day, and 1-year mortality among ICU-admitted HF patients. We further examined the contribution of SDOH, PF, and LS features to model performance and explored their potential as prognostic indicators to inform early clinical decision-making.

## Materials and methods

We performed a retrospective study using open-access databases including the Medical Information Mart for Intensive Care Database v1.4 (MIMIC-III, CareVue) and MIMIC-IV v1.0 (part), which were collected from the Beth Israel Deaconess Medical Center in Boston from 2001 to 2008 and 2008–2016, respectively [26,27]; An overview of the study flow is presented in Fig 1.

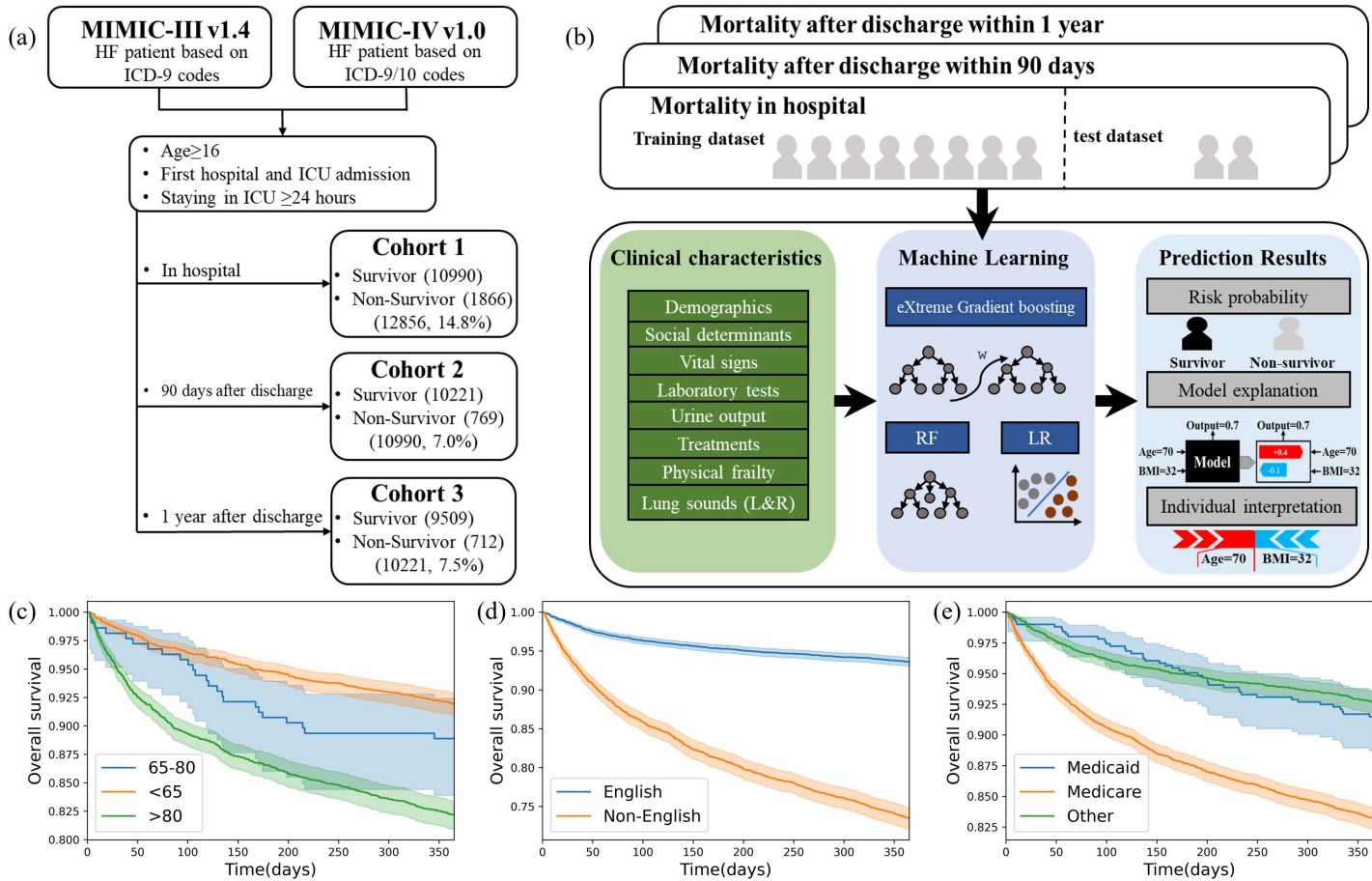

**Fig 1. Study design and survival outcomes across key subgroups.** (a) The study flows. (b) An overview of inclusion criteria with all study cohorts. (c)-(e) The K-M curves of age groups, primary language, and insurance.

## Study cohorts

We included all first-time ICU admissions for HF patients over 16 years old according to the International Classification of Diseases diagnostic codes [28]. We excluded the patients with unknown outcomes after discharge within 1 year and who stayed in the ICU for less than 24 hours. The inclusion criteria of the three study cohorts were displayed in Fig 1. Hospital survivors were used to analyze the out-of-hospital outcomes.

## Candidate variables

Data collected included the eight types of information for model development: Basic information of age, gender, weight, body mass index (BMI), Charlson Comorbidity Index (CCI), etc.; SDOH including primary language, insurance, marital status; Vital signs such as heart rate, respiratory rate, systolic blood pressure (SBP), Glasgow Coma Scale (GCS), etc.; Laboratory tests like glucose, creatinine, blood urea nitrogen (BUN), total bilirubin, etc.; Urine output; Treatments received like mechanical ventilation and vasopressors etc.; PF assessments including activity, fall risks, Braden nutrition and Braden activity; LS covering right and left sides. Details about all types of candidate variables are provided in the S1 File.

## Feature construction

The data measured during the first and last days of ICU admission were utilized to predict 90-day and 1-year mortality outcomes, while only first-day data were used for in-hospital mortality prediction. To distinguish between data measured on the first and last days, features from the last day were suffixed with '(leave)'. Additionally, we engineered new features to capture clinically relevant changes during the ICU stay, such as the difference in weight between the first and last day, which could reflect fluid balance and nutritional status, and the ratio of pulse oxygen saturation (SpO2) to fraction of inspired oxygen (FiO2), an indicator of respiratory efficiency.

Representative statistical features were calculated for each variable type to capture essential summary information. For example, we computed the maximum, minimum, mean, difference, and ratio values where appropriate, as these statistics can reflect the severity and variability of a patient's condition. To address missing data, the median value of each continuous variable was used for imputation, with the exception of FiO2, for which a value of 21% was imputed when missing, aligning with the standard atmospheric oxygen concentration. This approach ensures that the imputed values are both clinically plausible and minimize bias introduced by missing data. Furthermore, for variables with missing data in 30% or more of the patient population, we generated a missing value indicator to preserve information potentially embedded in the absence of data, as missingness itself may carry predictive value [29].

Overall, 79 features were constructed for in-hospital mortality prediction and 122 features for out-of-hospital mortality predictions. A comprehensive list of all variables studied, along with the specific features derived from them, is provided in S1 File. Detailed information on the missing data, including the missing ratios for each feature, can be found in S1 File. This thorough feature construction process ensures that the model captures a wide range of clinically relevant information while addressing potential issues related to missing data.

## Data preprocessing

To reduce the potential bias of ML algorithms in vulnerable subpopulations focusing on primary language and insurance characteristics, we adopted the reweighing algorithm before modeling, a data preprocessing technique, to compute the weights of patients to decrease the imbalance and unfairness [30,31].

## Model development

The eXtreme Gradient Boosting (XGBoost) algorithm was selected for developing the mortality prediction models for HF patients due to its superior performance in handling structured data and its ability to effectively manage missing values,

interactions, and non-linear relationships. XGBoost has consistently demonstrated high predictive accuracy in various healthcare applications, making it a robust choice for this study [32,33]. To provide a comprehensive evaluation, two other ML algorithms, logistic regression (LR) and random forests (RF), were used as baseline models. LR was chosen due to its simplicity, interpretability, and widespread use in clinical settings, while RF was selected for its strong performance in capturing complex interactions and non-linearities. By comparing the performance of XGBoost against these baseline models, we aim to illustrate the advantages of using advanced ML techniques in predicting both short- and long-term mortality outcomes in HF patients. For model development, all patients were randomly split into an 80% training set, which was used for training the models and tuning hyperparameters via grid search, and a 20% testing set, which was reserved for evaluating model performance.

### Model evaluation

The discrimination performance of our prediction models was assessed on the test set, comparing against the baseline models and conventional clinical scoring systems including SOFA and GWTG-HF score. Seven evaluation metrics were calculated with their 95% confidence intervals (95% CI), including the area under the curve of the receiver operating characteristic curve (AUROC), sensitivity, specificity, accuracy, F1 score, precision, and area under the precision recall curve.

### Model interpretation

The SHapley Additive exPlanations (SHAP) technique, which is based on a game theory framework, is a popular method for assessing the impact of a feature on model predictions and ranking its importance and relevance [34,35]. The Shapley value significantly enhances the interpretability of intricate ML models. We elected to utilize SHAP to elucidate the relative importance of diverse features that collectively influenced mortality prediction outcomes.

### Statistical analysis

The medians and interquartile ranges (IQR) for continuous variables were presented. The *t* test or Wilcoxon Rank Sum Test was used when appropriate to compare between survivors and non-survivors with HF. Categorical variables were reported by the total number and percentage. Two-sided *p*-values of less than.05 were considered statistically significant.

### Ethical statement

This study was exempt from institutional review board approval due to the retrospective design and lack of direct patient intervention. All data from patients were retrospectively collected from the electronic health care records systems (in the form of third-party public databases or hospital health care systems), which originated from daily clinical work.

All data were de-identified before the analysis. Third-party public databases were used in this study. The institutional review boards of the Massachusetts Institute of Technology (number 0403000206) and Beth Israel Deaconess Medical Center (number 2001-P-001699/14) approved the use of the database for research.

The requirement for individual patient consent was waived because the study did not impact clinical care, all protected health information was deidentified, and all available data in the databases were anonymous.

## Results

### Patient characteristics

The three study cohorts comprised 12,856 in-hospital patients (14.8% in-hospital mortality), 10,990 90-day discharged patients (7.0% 90-day mortality), and 10,990 1-year discharged patients (13.5% 1-year mortality), respectively. The characteristics of the aforementioned groups are presented in Table 1. In comparison to

**Table 1. The baseline characteristic of the total study cohorts divided by target outcomes.**

| | Survivor (n = 9,509) | In-hospital death (n = 1,866) | 90-day death (n = 769) | 1-year death (n = 1481) |
|---|---|---|---|---|
| **Basic information** | | | | |
| Age (y), (median, IQR) | 73.0 [63.0,83.0] | 79.0 [70.0,86.0] | 80.0 [72.0,86.0] | 78.0 [70.0,85.0] |
| Female, (%) | 4367 (45.9) | 903 (48.4) | 345 (44.9) | 687 (46.4) |
| BMI[a] (kg/m²), (median, IQR) | 28.2 [24.6,33.3] | 26.5 [23.0,31.5] | 25.8 [22.4,29.6] | 26.1 [22.5,30.2] |
| CCI[a] score, (median, IQR) | 7.0 [5.0,8.0] | 7.0 [6.0,9.0] | 7.0 [6.0,9.0] | 7.0 [6.0,9.0] |
| Ethnicity (%) | | | | |
| Asian | 194 (2.0) | 37 (2.0) | 10 (1.3) | 26 (1.8) |
| Black | 893 (9.4) | 120 (6.4) | 59 (7.7) | 128 (8.6) |
| Hispanic | 248 (2.6) | 36 (1.9) | 6 (0.8) | 19 (1.3) |
| Other/Unknown | 1387 (14.6) | 387 (20.7) | 124 (16.1) | 235 (15.9) |
| White | 6787 (71.4) | 1286 (68.9) | 570 (74.1) | 1073 (72.5) |
| **PF** | | | | |
| Fall risk (%) | 2756 (29.0) | 674 (36.1) | 415 (54.0) | 801 (54.1) |
| Activity (%) | | | | |
| Bed | 6538 (69.0) | 1645 (88.8) | 630 (82.2) | 1191 (80.7) |
| Sit | 1944 (20.5) | 147 (7.9) | 93 (12.1) | 193 (13.1) |
| Stand | 987 (10.4) | 61 (3.3) | 43 (5.6) | 91 (6.2) |
| Braden nutrition (%) | | | | |
| Adequate/ Excellent | 3588 (41.0) | 309 (16.7) | 208 (27.5) | 438 (30.1) |
| Probably Inadequate | 4990 (53.0) | 1202 (65.1) | 478 (63.2) | 902 (61.9) |
| Very Poor | 568 (6.0) | 336 (18.2) | 70 (9.3) | 117 (8.0) |
| Braden mobility (%) | | | | |
| Slight Limitations | 5928 (63.0) | 601 (32.2) | 359 (47.5) | 712 (48.8) |
| Very Limited | 3486 (37.0) | 1246 (66.8) | 397 (52.5) | 746 (51.2) |
| Braden activity (%) | | | | |
| Bedfast | 7232 (76.8) | 1680 (91.0) | 647 (85.6) | 1244 (85.4) |
| Chairfast | 1385 (14.7) | 110 (6.0) | 72 (9.5) | 136 (9.3) |
| Walks Frequently | 796 (8,4) | 57 (3.1) | 37 (4.9) | 77 (5.3) |
| **Right upper lobe LS[a] (%)** | | | | |
| Clear | 7188 (75.7) | 1061 (56.9) | 510 (66.3) | 1018 (68.8) |
| Crackles | 155 (1.6) | 56 (3.0) | 16 (2.1) | 24 (1.6) |
| Diminished | 706 (7.4) | 176 (9.4) | 50 (6.5) | 84 (5.7) |
| Stridor/ Pleural Friction/ etc. | 1451 (15.3) | 572 (30.7) | 193 (25.1) | 354 (23.9) |
| **SDOH** | | | | |
| Primary language (%) | | | | |
| English | 6657 (70.0) | 970 (52.0) | 247 (32.1) | 453 (30.6) |
| Non-English | 2852 (29.9) | 896 (48.0) | 522 (67.9) | 116 (7.8) |
| Insurance (%) | | | | |
| Medicaid | 463 (4.9) | 61 (3.3) | 10 (1.3) | 43 (2.9) |
| Medicare | 5919 (62.2) | 1371 (73.5) | 637 (82.8) | 1193 (80.6) |
| Other | 3127 (32.9) | 434 (23.3) | 122 (15.9) | 245 (16.5) |
| **Outcome** | | | | |
| Days before ICU admission (median, IQR) | 0.1 [0.0,1.0] | 0.1 [0.0,1.4] | 0.1 [0.0,1.7] | 0.1 [0.0,1.5] |
| Days of ICU admission, (median, IQR) | 2.8 [1.8,4.9] | 4.6 [2.3,8.9] | 3.6 [2.1,7.0] | 3.3 [2.0,6.2] |
| Days of hospital admission, (median, IQR) | 8.7 [5.7,13.9] | 8.7 [4.5,15.8] | 11.9 [7.0,18.8] | 11.2 [6.8,18.0] |

survivors, patients who experienced unfavorable outcomes demonstrated a higher prevalence of advanced age, lower BMI, a greater proportion of urgent admissions, elevated CCI scores, elevated fall risk, a predominantly bedridden status, poor nutritional status, markedly limited mobility, muddy LS, a higher proportion of non-English primary languages, and Medicare insurance coverage. Additionally, these patients exhibited prolonged ICU and hospital stay durations. Furthermore, a consistent observation was made of a decrease in weight, limited activity, increased fall risk, increased poor malnutrition, muddy LS, Medicare, and limited language speaking when comparing each term of bad outcome (see S1 File). However, higher mechanical ventilation was only observed in hospital non-survivors.

## Model performance evaluation

The performance of the models was evaluated on the test sets using AUROC curves, as illustrated in Fig 2. The discrimination of the prediction models demonstrated satisfactory performance, as evidenced by the AUROC values, which were 0.836 (95% CI: 0.831–0.844) for in-hospital mortality, 0.790 (95% CI: 0.780–0.800) for 90-day mortality, and 0.789 (95% CI: 0.780–0.799) for 1-year mortality. The AUROC for short-term outcome prediction was relatively higher than that for

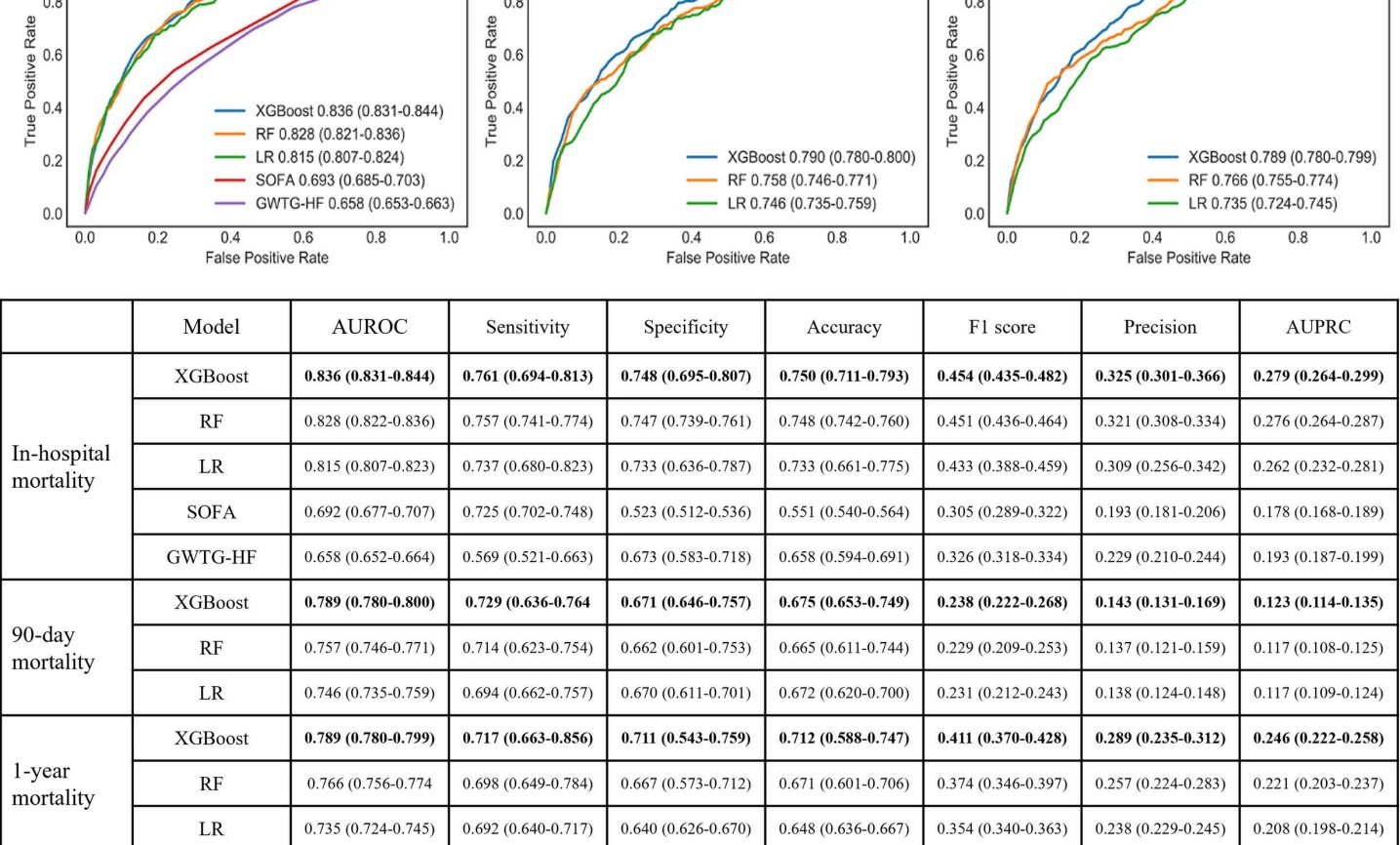

| | Model | AUROC | Sensitivity | Specificity | Accuracy | F1 score | Precision | AUPRC |
|---|---|---|---|---|---|---|---|---|
| In-hospital mortality | XGBoost | **0.836 (0.831-0.844)** | **0.761 (0.694-0.813)** | **0.748 (0.695-0.807)** | **0.750 (0.711-0.793)** | **0.454 (0.435-0.482)** | **0.325 (0.301-0.366)** | **0.279 (0.264-0.299)** |
| | RF | 0.828 (0.822-0.836) | 0.757 (0.741-0.774) | 0.747 (0.739-0.761) | 0.748 (0.742-0.760) | 0.451 (0.436-0.464) | 0.321 (0.308-0.334) | 0.276 (0.264-0.287) |
| | LR | 0.815 (0.807-0.823) | 0.737 (0.680-0.823) | 0.733 (0.636-0.787) | 0.733 (0.661-0.775) | 0.433 (0.388-0.459) | 0.309 (0.256-0.342) | 0.262 (0.232-0.281) |
| | SOFA | 0.692 (0.677-0.707) | 0.725 (0.702-0.748) | 0.523 (0.512-0.536) | 0.551 (0.540-0.564) | 0.305 (0.289-0.322) | 0.193 (0.181-0.206) | 0.178 (0.168-0.189) |
| | GWTG-HF | 0.658 (0.652-0.664) | 0.569 (0.521-0.663) | 0.673 (0.583-0.718) | 0.658 (0.594-0.691) | 0.326 (0.318-0.334) | 0.229 (0.210-0.244) | 0.193 (0.187-0.199) |
| 90-day mortality | XGBoost | **0.789 (0.780-0.800)** | **0.729 (0.636-0.764)** | **0.671 (0.646-0.757)** | **0.675 (0.653-0.749)** | **0.238 (0.222-0.268)** | **0.143 (0.131-0.169)** | **0.123 (0.114-0.135)** |
| | RF | 0.757 (0.746-0.771) | 0.714 (0.623-0.754) | 0.662 (0.601-0.753) | 0.665 (0.611-0.744) | 0.229 (0.209-0.253) | 0.137 (0.121-0.159) | 0.117 (0.108-0.125) |
| | LR | 0.746 (0.735-0.759) | 0.694 (0.662-0.757) | 0.670 (0.611-0.701) | 0.672 (0.620-0.700) | 0.231 (0.212-0.243) | 0.138 (0.124-0.148) | 0.117 (0.109-0.124) |
| 1-year mortality | XGBoost | **0.789 (0.780-0.799)** | **0.717 (0.663-0.856)** | **0.711 (0.543-0.759)** | **0.712 (0.588-0.747)** | **0.411 (0.370-0.428)** | **0.289 (0.235-0.312)** | **0.246 (0.222-0.258)** |
| | RF | 0.766 (0.756-0.774) | 0.698 (0.649-0.784) | 0.667 (0.573-0.712) | 0.671 (0.601-0.706) | 0.374 (0.346-0.397) | 0.257 (0.224-0.283) | 0.221 (0.203-0.237) |
| | LR | 0.735 (0.724-0.745) | 0.692 (0.640-0.717) | 0.640 (0.626-0.670) | 0.648 (0.636-0.667) | 0.354 (0.340-0.363) | 0.238 (0.229-0.245) | 0.208 (0.198-0.214) |

**Fig 2. The comparison of the three final models against 2 baseline ML models and conventional clinical scores.** (a) in-hospital, (b) 90-day and (c) 1-year.

long-term prediction. Furthermore, our model was compared against two baseline ML models and conventional clinical scores for all outcomes of interest, with a total of seven metrics considered in detail. As shown, our prediction models consistently demonstrated superior performance compared to all other models and scores.

### Feature importance and interpretation

In Fig 3, we present the top 20 risk factors for in-hospital, 90-day, and 1-year mortality predictions, with all feature rankings provided in S1 File. The relative importance of features is indicated by their position on the y-axis, with the most influential features positioned higher. The x-axis represents the SHAP value, which reflects the impact of each feature on the model's predictions. A positive SHAP value indicates that the feature contributes to an elevated risk of mortality. In the case of continuous features, a color gradient from red to blue is used to indicate a decrease in feature value, with red representing a higher value and blue representing a lower value. In the case of binary features, the color red indicates the presence of a condition (e.g., "yes"), while blue represents its absence ("no").

As illustrated in Fig 3, several factors, including the GCS, urine output, BUN, age, respiratory rate, SBP, and weight, were identified as crucial for the early assessment of in-hospital mortality. It is noteworthy that Braden mobility and nutrition, activity, and right upper lobe LS emerged as novel and significant predictors, ranked 3rd, 11th, 12th, and 18th, respectively. These findings indicate that integrating functional and nutritional status with traditional vital signs may facilitate enhanced early risk stratification in clinical practice.

In Fig 3, the pre-ICU discharge status indicators, including BUN, temperature, urine output, SBP, respiratory rate, and weight, exhibited a more pronounced correlation with the 90-day outcomes. Age and fall risk (leave) were the most significant predictors, indicating that elderly patients with impaired mobility or a higher fall risk at discharge are at an elevated risk for adverse outcomes. The prominence of primary language and insurance status (SDOH) ranked 4th and 11th, respectively, underscoring the necessity of considering social factors in post-discharge care planning. This could facilitate the implementation of tailored interventions and follow-up strategies.

Our findings indicate that a combination of basic demographic information, admission, and discharge status in the ICU is a significant predictor of one-year outcomes in Fig 3. Factors such as primary language, CCI score, age, BMI, pre-ICU length of stay (LOS), and insurance status were identified as the top 13 predictors. Furthermore, creatinine, chloride, and platelet levels were identified as significant biomarkers, ranking 9th, 17th, and 20th, respectively. The assessment of fall risk prior to discharge was identified as the most critical predictor, ranking 1st, emphasizing the importance of comprehensive discharge assessments to predict long-term outcomes.

In order to establish a connection between model explanation and clinical application, we presented the practical utility of SHAP values in S1 File through the use of two case studies, one survivor and one non-survivor, for each outcome prediction. The case studies demonstrate how SHAP values can be employed to identify pivotal risk factors for individual patients, thereby facilitating the formulation of personalized treatment plans. For example, a high SHAP value for BUN or fall risk could prompt closer monitoring or early intervention to mitigate the identified risks. By translating model explanations into actionable insights, clinicians can enhance decision-making and patient outcomes through more targeted and informed care strategies.

### Contribution of SDOH, PF, and LS

The performance contribution of SDOH (insurance and primary language), PF (fall risk, activity, Braden nutrition, Braden mobility, Braden activity) and LS were separately evaluated by removing them to re-train all outcome prediction models. In Fig 4 and S1 File displayed the AUROC comparisons with the fully models. Our findings indicated that all of the factors could enhance the discrimination, and it was obvious in 90-day and 1-year outcome prediction.

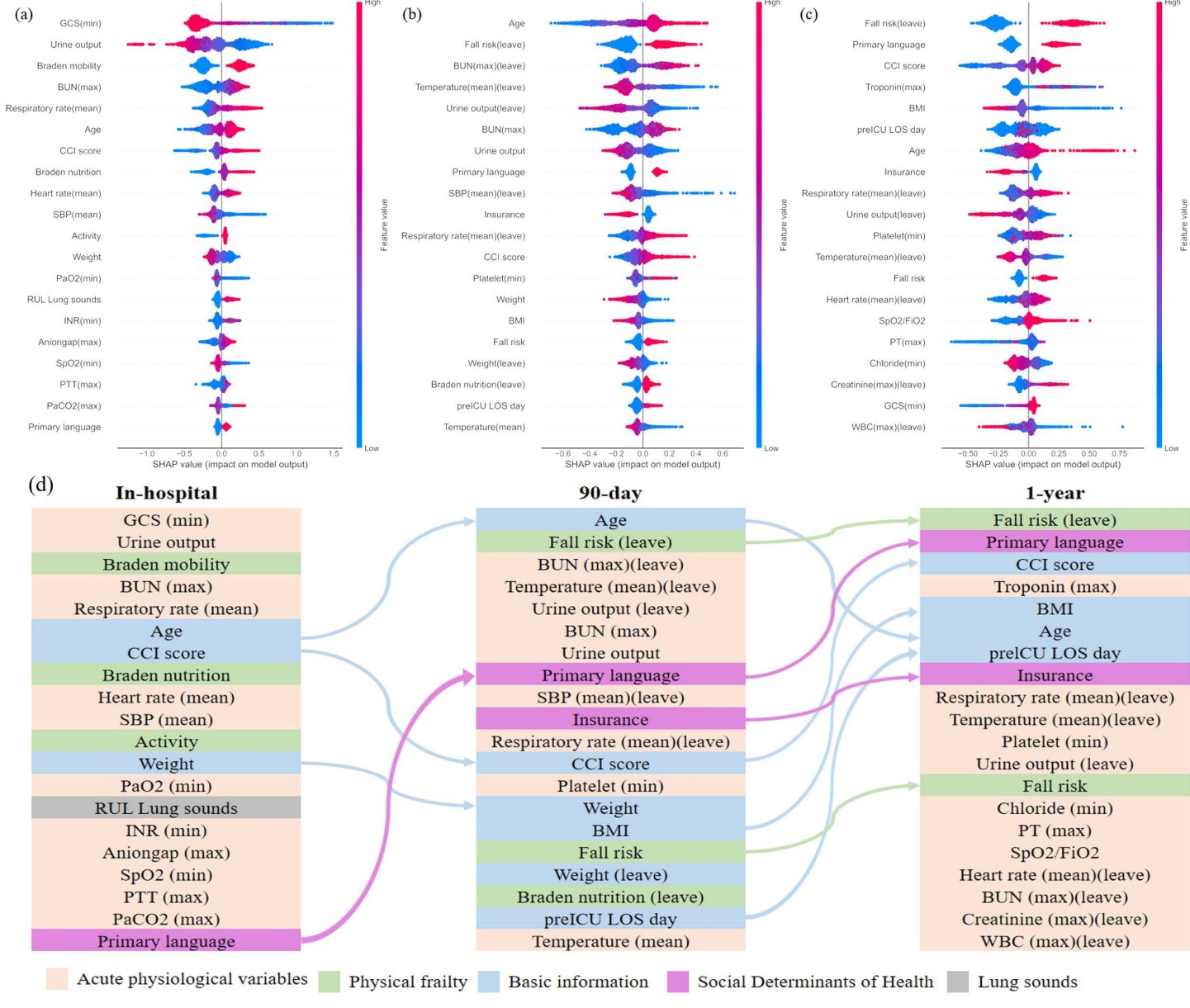

**Fig 3. The models' interpretation of the top 20 important features based on the optimal models.** (a) in-hospital mortality. (b) 90-day mortality. (c) 1-year mortality. The higher the SHAP value a feature is given, the higher the risk of death for the patient. The red part in feature value represents a higher value. (d) Alluvial plot of the top 20 risk factors for outcome predictions of HF patients.

## Sensitivity analysis of different variables sets

We performed two types of sensitivity analysis to acquire the stability models. The top 5 to all variables (interval 5) were used for model building to analyze the change of discrimination in all outcome predictions. S1 File present the variation of AUROC in the development and test sets for three tasks. For the short-term prediction, AUROC over 0.8 when 40 features, importance ranked, were included. For the long-term predictions, the models can achieve relatively acceptable predictive performance when including 30 ranked features. Furthermore, the model with only 5 features outperformed both clinical

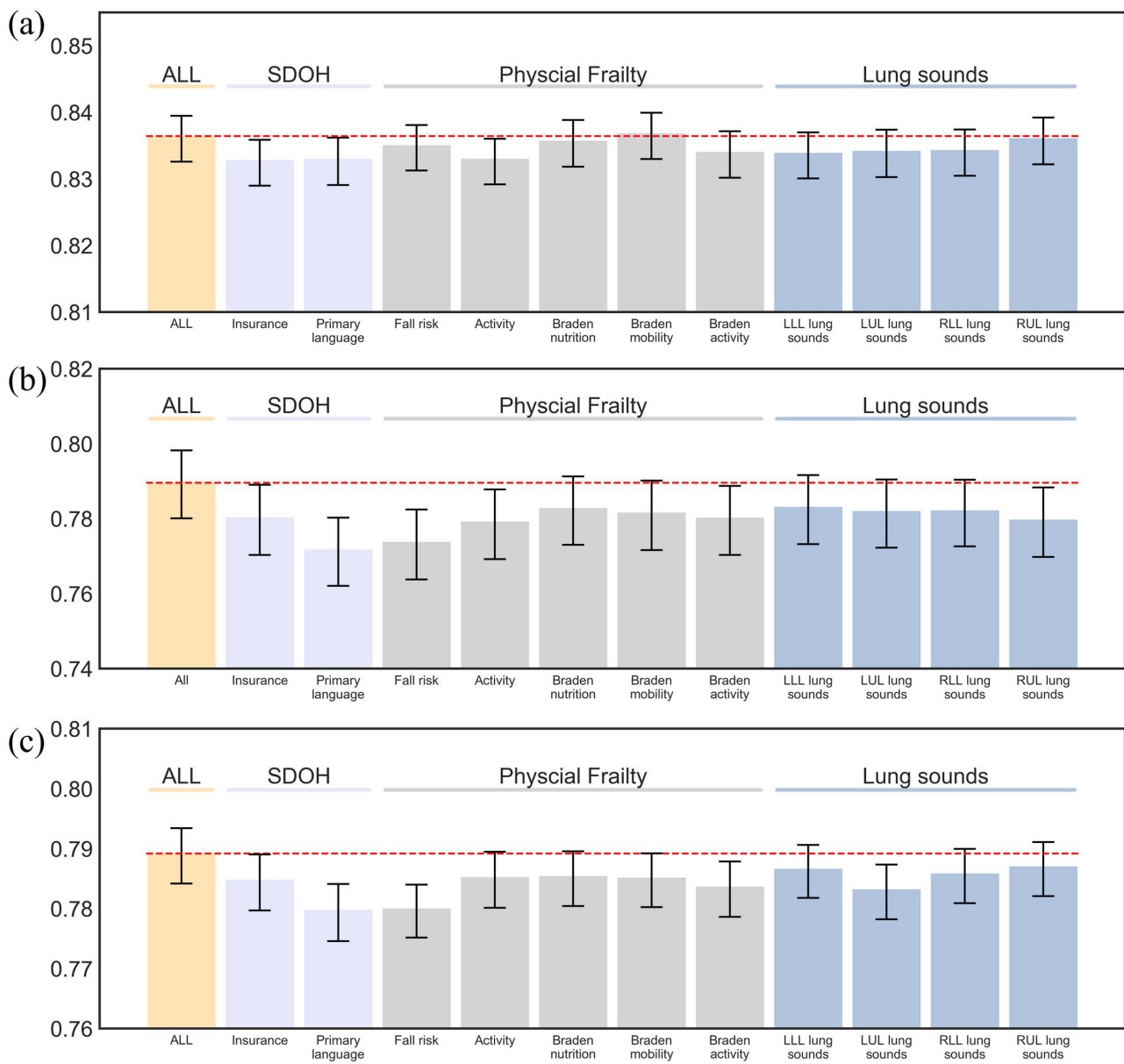

**Fig 4. The reduction of discrimination when dropping the variables of SDOH, PF and LS.** (a) in-hospital mortality. (b) 90-day mortality. (c) 1-year mortality.

scores in in-hospital mortality prediction. The importance of assessing HF patients on their first and last days in ICU for long-term prognosis was analyzed separately, shown in S1 File. Out-of-hospital outcomes can be better evaluated by the status assessment of patients before ICU discharge than admission. Combining the two parts allows for more precise prediction.

## Discussion

This retrospective prognostic study developed and validated three ML models to predict in-hospital, 90-day, and 1-year mortality in critically ill patients with HF, based on data from the first and last ICU admission days. To enhance model

performance and mitigate disparities, we incorporated key variables including SDOH such as primary language and insurance type; PF indicators including activity level, fall risk, Braden mobility, and nutrition scores; and LS. Our findings demonstrate that these features significantly improved the predictive accuracy of the models, particularly in long-term outcome assessments.

We observed that different variables contributed uniquely to short- versus long-term prognostication. Braden mobility, nutrition, and activity were more influential in predicting in-hospital outcomes, whereas fall risk emerged as a stronger predictor for 90-day and 1-year mortality. The inclusion of SDOH became increasingly important in out-of-hospital outcome prediction. Across all timepoints, our models consistently outperformed baseline ML models and widely used clinical scores such as SOFA and GWTG-HF, achieving AUROCs of 0.836 (in-hospital), 0.790 (90-day), and 0.789 (1-year).

To evaluate generalizability, we conducted an external validation using the eICU Collaborative Research Database, which includes ICU admissions from over 200 U.S. hospitals. Although follow-up data for 90-day and 1-year outcomes were not available, the model performed well in predicting in-hospital mortality, with an AUROC of 0.769 (see S1 File).

A novel contribution of our study is the inclusion of primary language as a predictor variable—an aspect rarely considered in previous HF prognostic models. Language proficiency ranked as the 13th, 4th, and 2nd most important feature for in-hospital, 90-day, and 1-year mortality, respectively (Fig 3). Kaplan-Meier curves revealed substantial survival differences between English-proficient and non-proficient patients (Fig 1), underscoring the need to incorporate language into risk stratification or to examine it as a separate analytic factor.

Similarly, we found that insurance type—particularly Medicare, which predominantly covers older adults and those with disabilities—was associated with long-term outcomes. Patients covered by Medicare showed lower survival rates, reflecting both age-related and socioeconomic vulnerabilities (Fig 1). While prior studies have examined the link between insurance and HF outcomes [41, 42], our study is among the first to evaluate insurance as a formal input to prediction models. Its inclusion ranked within the top 13 features (Fig 3), and model performance declined when the insurance variable was removed (see S1 File), reinforcing its prognostic value.

We further assessed PF by analyzing Braden scale components and activity status. Braden mobility, nutrition, and activity were significant predictors of in-hospital outcomes, reflecting the patient's immediate functional capacity. In contrast, fall risk at ICU discharge played a more critical role in long-term outcome prediction, suggesting its potential utility for post-discharge care planning and rehabilitation.

Several conventional clinical variables, including age, BUN, and the CCI score, consistently ranked as top predictors across all models, reinforcing their clinical relevance. Notably, urine output on the last ICU day ranked 6th in importance for long-term outcomes, indicating its role as a potential marker of renal function and volume status. Conversely, right upper lobe LS emerged as a relevant predictor primarily for in-hospital mortality (ranked 18th), likely due to its association with acute pulmonary complications rather than long-term prognosis.

Our work contributes several key advances to the field of HF outcome prediction. We demonstrated that integrating SDOH—specifically primary language and insurance type—can enhance long-term risk assessment. We also expanded the consideration of frailty metrics, including fall risk, which proved to be a meaningful predictor of post-discharge mortality. By selecting variables readily available at ICU admission and discharge, we developed robust, explainable ML models that outperform existing tools and support early identification of high-risk HF patients both during hospitalization and after discharge.

## Study limitations

Several limitations should be acknowledged in this study. First, the retrospective nature of the analysis and the use of data from a single institution may limit the generalizability of the findings to broader HF populations and diverse clinical settings. Second, the prediction models were developed using data only from the first and last ICU days, excluding the intermediate course of ICU treatment. This omission may have led to a loss of clinically relevant information, potentially

affecting model performance. Third, in order to simplify feature extraction, we represented dynamic variables using summary statistics such as maximum, minimum, or mean values over specified time intervals. While this approach facilitated model development, it disregarded the temporal patterns and variability inherent in time-series data, which may contain important prognostic information. Incorporating full time-series dynamics will be a focus of future research.

## Conclusion

In this prognostic study, we developed and validated three outcome prediction models for patients with HF, targeting in-hospital, 90-day, and 1-year mortality. The models integrated a comprehensive set of predictors, including demographic and clinical variables (age, CCI score, BMI), PF indicators (Braden mobility and nutrition, activity level, and fall risk), SDOH (primary language and insurance status), vital signs (GCS, systolic blood pressure, respiratory rate, and temperature), laboratory values (BUN), urine output, and LS. These variables collectively enabled more accurate and individualized risk stratification for both short- and long-term outcomes. The models demonstrated strong discriminatory power and interpretability, offering promising tools to support clinical decision-making and optimize discharge planning for patients with HF.

## Supporting information

**S1 File. Supplementary materials: cohort characteristics, detailed model performance, and feature analysis for mortality risk prediction.**
(DOCX)

## Author contributions

**Conceptualization:** Zhenyue Gao, Xiaoli Liu, Yumeng Peng, Muyang Yan.

**Data curation:** Zhenyue Gao, Xiaoli Liu, Yu Kang, Xiu Zhang, Mengwei Li.

**Formal analysis:** Xiaoli Liu, Yu Kang, Mengwei Li, Muyang Yan.

**Funding acquisition:** Zhengbo Zhang.

**Investigation:** Pan Hu, Xiu Zhang, Yumeng Peng, Pengming Yu, Zhengbo Zhang.

**Methodology:** Zhenyue Gao, Wei Yan, Wendong Xiao.

**Resources:** Pan Hu, Zhengbo Zhang.

**Software:** Wei Yan.

**Supervision:** Pan Hu, Wei Yan, Pengming Yu, Qing Zhang.

**Validation:** Yumeng Peng, Qing Zhang, Wendong Xiao.

**Visualization:** Yumeng Peng.

**Writing – original draft:** Zhengbo Zhang, Wendong Xiao.

**Writing – review & editing:** Zhengbo Zhang, Wendong Xiao.

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
