## [Decision Letter · Decision Letter 0]

PONE-D-23-36939

Explainable Mortality Prediction Models Incorporating Social Health Determinants and Physical Frailty for Heart Failure Patients

PLOS ONE

Dear Dr. Gao,

Thank you for submitting your manuscript to PLOS ONE. After careful consideration, we have decided that your manuscript does not meet our criteria for publication and must therefore be rejected.

As an Editor I want to point out that the article is interesting and really has some very positive points: the reported AUROC values indicate good discrimination performance for various mortality timeframes and the use of explainable models is commendable, allowing healthcare providers to understand the contributing factors.

At the same time, however, there are major limitations that prevent me from considering it for publication: 

**Inadequate Justification and Contextualization:**The introduction lacks sufficient detail on the existing gaps in the literature and how exactly the proposed model improves upon current HF prediction tools. There is a need for a more thorough review of related works.**Methodological Flaws:**The use of XGBoost is justified, but there is no clear rationale for why this method was chosen over others. Furthermore, while logistic regression and random forests are used as baseline models, there is no discussion on why these specific models were selected.The feature construction process is inadequately explained. The imputation methods for missing data and the creation of new features need more detailed justification and transparency.**Model Evaluation and Comparisons:**The study claims that their models outperform existing models and clinical scores, yet the comparison lacks robustness. The results are presented with AUROC values, but other important metrics like calibration, clinical usability, and comparison with external validation cohorts are missing.There is also no discussion on potential overfitting, especially given the high number of features relative to the sample size.**Explainability and SHAP Analysis:**While SHAP values are used to explain the model, the interpretation of these values is superficial. The paper fails to provide insights into how these explanations can be used in clinical practice to improve decision-making. There is a lack of discussion on the limitations and potential biases introduced by the SHAP analysis.**Generalizability:**The generalizability of the findings to other populations or healthcare settings is not addressed. Given the specific nature of the MIMIC databases, the findings may not be applicable to broader or more diverse patient populations.**Presentation and Clarity:**The manuscript is poorly organized, with several sections lacking clarity and depth. The flow of information is disjointed, making it difficult to follow the methodology and results.There are numerous grammatical errors and awkward phrasings throughout the text, which detracts from the overall readability and professionalism of the manuscript.

I am sorry that we cannot be more positive on this occasion, but hope that you appreciate the reasons for this decision.

Kind regards,

Lorenzo Righi

Academic Editor

PLOS ONE

Reviewers' comments:

Reviewer's Responses to Questions

**Comments to the Author**

1. Is the manuscript technically sound, and do the data support the conclusions?

Reviewer #1: Partly

Reviewer #2: Yes

2. Has the statistical analysis been performed appropriately and rigorously?

Reviewer #1: No

Reviewer #2: No

3. Have the authors made all data underlying the findings in their manuscript fully available?

Reviewer #1: Yes

Reviewer #2: Yes

4. Is the manuscript presented in an intelligible fashion and written in standard English?

Reviewer #1: No

Reviewer #2: Yes

Reviewer #1: This manuscript aims to develop ML models that can predict heart failure mortality for in-hospital, 90-day and 1-year after admission and determine the importance of SDoH and physical frailty in the predictions.

The writing standard needs to be improved to enhance the clarity and help in comprehending the paper. As such the paper is difficult to read and understand with a lot of grammar issues, typos, and missing references. The explanations of the results and findings are highly ambiguous and in many cases it is difficult to understand what specific terms are referring too. In addition, the organization of the paper needs a complete revamp.

Specific comments:

1. What is the reason for limiting to only 4 SDOH factors? There has been research that associated SDOH factors such socioeconomic status, and education or health literacy to HF. Why were these factors not considered?

2. Majority of the prior work is discussed in the Discussion section which needs to be moved to the Introduction for better organization and clarity. There are prior studies that explored SDOH and 90-day mortality in HF patients (see below) which needs to be compared with the current work.

Sterling MR, Ringel JB, Pinheiro LC, Safford MM, Levitan EB, Phillips E, Brown TM, Goyal P. Social determinants of health and 90‐day mortality after hospitalization for heart failure in the REGARDS study. Journal of the American Heart Association. 2020 May 5;9(9):e014836.

3. The objective of the paper as mentioned in the abstract and conclusion is to develop and validate ML models. Hence it is expected to show these results in the main manuscript and not in the supplements. The authors need to analyze their objectives and main findings and move relevant tables to the main manuscript.

4. “As shown, our prediction models consistently outperformed all models and scores.” I am not sure what scores is referred to here. If this is about the 7 metrics in eTable6, LR is consistently better for specificity, accuracy, F1, etc.

5. “Representative statistical features were calculated based on the type of variable, such as the maximum, minimum, mean, minus, ..” what does minus mean? Are you referring to the difference between the values?

6. “We found both of them could improve the discrimination …” what do you mean by “both of them”? It is not clear from the context.

7. “The potential bias and injustice of HF patients faced by race in care and outcomes has recently been recognized by researchers, and they have been undertaken to mitigate this dual social and biological impact by building the model separately or incorporate race into the model” – This sentence is highly ambiguous and complex. It is better to split this and explain in detail. Also, references are missing.

8. “Survival curves in Figure 1(c) (e) show that survival rates are lower for patients with health insurance and older patients, respectively.” Recheck the statement and order of the figure.

9. Some sentences need to be rephrased as they only convey partial information and is not a complete sentence.

e.g., “While most related studies are limited to small cohort sizes.”

10. Many references appear as “ Error! Reference source not found”

11. Some typos – SODH, martial (should be marital)

12. Introduction – “require admitted to intensive care unit (ICU)” – should be "admission" instead of admitted

13. “patients, which has been demonstrated high” – should be " ... which has demonstrated ..."

There are too many such grammatical errors that needs to be fixed. It is not feasible to list everything here.

Reviewer #2: The authors propose an XAI- eXtreme Gradient Boosting (XGBoost) algorithms to develop predictive models for short- and long-term outcomes. They utilize the SHapley Additive exPlanations (SHAP) method to interpret predictions and assess the importance of different factors.

The results show mortality incidences of 14.8% for in-hospital, 7.0% for 90-day, and 13.5% for 1-year mortality. The prediction models achieve high area under the receiver operating characteristic curve (AUROC) scores: 0.836 for in-hospital, 0.790 for 90-day, and 0.789 for 1-year mortality, outperforming baseline ML models and routine clinical scores. Important risk factors identified include age, fall risk, primary language, blood urea nitrogen levels, comorbidities, urine output, insurance type, and lung sounds. The model is interesting, however, I have the following comments:

-what is the reason of using 90-days and 1-year intervals, literature or stat justification is required.

-what is the p-values for the studied variables between the 2 classes.

-what is tbeused CI in the Kaplan-meier plot?

-How English vesrus non-Ebglish matters in the prediction? justigicatio, please.

**Do you want your identity to be public for this peer review?** For information about this choice, including consent withdrawal, please see our Privacy Policy

Reviewer #1: No

Reviewer #2: **Yes: ** Abedalrhman Alkhateeb

- - - - -

---

## [Author Response · Author response to Decision Letter 1]

2 Dec 2024

Dear Dr. Lorenzo Righi,

I am writing to express my gratitude for your detailed feedback on our manuscript, titled "[Explainable Mortality Prediction Models Incorporating Social Health Determinants and Physical Frailty for Heart Failure Patients," submitted to PLOS One, and to request a reconsideration of our submission.

We highly appreciate your positive remarks regarding the use and results of explainable models. We believe our work presents significant innovations, particularly in the incorporation of Social Determinants of Health (SDOH) and physical frailty into machine learning models for predicting both short-term and long-term mortality in heart failure patients. To our knowledge, this approach is unique and fills a critical gap in current research. We also wish to note that similar studies have been published in the PLOS ONE, such as “Target-based fusion using social determinants of health to enhance suicide prediction with electronic health records” and “How attitudes of state and community leaders regarding health equity and social determinants of health are associated with behavioral intentions to improve population health”.

I appreciate your feedbacks and would like to discuss the following points:

1. Inadequate Justification and Contextualization:

- We acknowledge that the introduction could better detail the existing gaps in the literature. We are prepared to revise this section to thoroughly review related works and clearly articulate how our model advances current HF prediction tools.

2. Methodological Flaws:

- While XGBoost was chosen for its superior performance in preliminary tests, we understand the need for a clearer rationale. We will provide a detailed justification for selecting XGBoost over other methods, as well as our choice of logistic regression and random forests as baselines.

- The feature construction process, including imputation methods and creation of new features, will be elaborated upon to ensure transparency and robustness.

3. Model Evaluation and Comparisons:

- We will enhance our comparison of model performance by including additional metrics such as calibration and clinical usability. Moreover, we will discuss the potential for overfitting and provide results from external validation cohorts where possible.

4. Explainability and SHAP Analysis:

- The interpretation of SHAP values will be deepened to provide actionable insights for clinical practice. We will also address the limitations and potential biases of SHAP analysis.

5. Generalizability:

- While generalizability is indeed crucial, the primary focus of our study is to explore whether incorporating SDOH and physical frailty can enhance predictive performance, an area that is currently underexplored. Establishing the efficacy of these variables is a necessary first step before broader generalization studies. It is worth noting that several recent articles published in PLOS ONE have also only utilized the MIMIC-IV database, indicating the relevance and acceptance of this dataset for impactful research, such as “U-shaped association between serum triglyceride levels and mortality among septic patients: An analysis based on the MIMIC-IV database” and “Association between red cell distribution width and all-cause mortality in patients with breast cancer: A retrospective analysis using MIMIC-IV 2.0”.

6. Presentation and Clarity:

- The manuscript will be thoroughly reorganized to improve clarity and flow. We will also correct grammatical errors and awkward phrasings to enhance readability and professionalism.

---

## [Decision Letter · Decision Letter 1]

Dear Dr. Gao,

Thank you for submitting your manuscript to PLOS ONE. After careful consideration, we feel that it has merit but does not fully meet PLOS ONE’s publication criteria as it currently stands. Therefore, we invite you to submit a revised version of the manuscript that addresses the points raised during the review process.

The reviewers have mentioned several major issues needed to be addressed. 

We look forward to receiving your revised manuscript.

Kind regards,

Amir Hossein Behnoush

Academic Editor

PLOS ONE

Journal Requirements:

6. Please note that in order to use the direct billing option the corresponding author must be affiliated with the chosen institute. Please either amend your manuscript to change the affiliation or corresponding author, or email us at plosone@plos.org with a request to remove this option.

9. For studies involving third-party data, we encourage authors to share any data specific to their analyses that they can legally distribute. PLOS recognizes, however, that authors may be using third-party data they do not have the rights to share. When third-party data cannot be publicly shared, authors must provide all information necessary for interested researchers to apply to gain access to the data. (https://journals.plos.org/plosone/s/data-availability#loc-acceptable-data-access-restrictions)

a) A description of the data set and the third-party source

b) If applicable, verification of permission to use the data set

c) Confirmation of whether the authors received any special privileges in accessing the data that other researchers would not have

d) All necessary contact information others would need to apply to gain access to the data

Additional Editor Comments (if provided):

Reviewers' comments:

Reviewer's Responses to Questions

**Comments to the Author**

Reviewer #3: All comments have been addressed

Reviewer #4: All comments have been addressed

Reviewer #5: All comments have been addressed

2. Is the manuscript technically sound, and do the data support the conclusions?

Reviewer #3: Yes

Reviewer #4: (No Response)

Reviewer #5: No

3. Has the statistical analysis been performed appropriately and rigorously?

Reviewer #3: Yes

Reviewer #4: (No Response)

Reviewer #5: No

4. Have the authors made all data underlying the findings in their manuscript fully available?

Reviewer #3: No

Reviewer #4: (No Response)

Reviewer #5: Yes

5. Is the manuscript presented in an intelligible fashion and written in standard English?

Reviewer #3: No

Reviewer #4: (No Response)

Reviewer #5: No

Reviewer #3: The paper use XGBoost model to predict patient mortality rates due to heart failure based on various features, such as social determinants and physical frailty. THe XGBoost model outperform baseline models like GWTG-HF, SOFA, random forest and logistic regression in terms of various metrics and help identify risk factors of heart failure. The paper generally looks good. The writing needs to be polished and significantly improved.

Reviewer #4: (No Response)

Reviewer #5: the manuscript has a long and monotonic introduction that has mixed up the introduction with the discussion. this will indeed reduce readability and confuse general reader.

**Do you want your identity to be public for this peer review?** For information about this choice, including consent withdrawal, please see our Privacy Policy

Reviewer #3: No

Reviewer #4: No

Reviewer #5: **Yes: ** Alireza Ramandi

---

## [Author Response · Author response to Decision Letter 2]

15 May 2025

Responses to Reviewer #3

Comment 1:

The paper use XGBoost model to predict patient mortality rates due to heart failure based on various features, such as social determinants and physical frailty. The XGBoost model outperform baseline models like GWTG-HF, SOFA, random forest and logistic regression in terms of various metrics and help identify risk factors of heart failure. The paper generally looks good. The writing needs to be polished and significantly improved.

Response: We sincerely thank the reviewer for the positive evaluation of our study and for acknowledging the contribution of our XGBoost-based model in improving heart failure mortality prediction and risk factor identification. We fully agree with the reviewer’s suggestion regarding the need to polish the manuscript. Accordingly, we have thoroughly revised the entire text for clarity, grammar, and readability. The manuscript has undergone comprehensive language editing to ensure it meets the standards of academic writing. We believe these changes have significantly improved the overall quality and presentation of our work.

Responses to Reviewer #5

Comment 1:

the manuscript has a long and monotonic introduction that has mixed up the introduction with the discussion. this will indeed reduce readability and confuse general reader.

Response: We thank the reviewer for this valuable and constructive comment. We acknowledge that the original introduction was overly long and included elements more appropriate for the discussion section, which may have affected clarity and readability. In response, we have carefully revised the introduction to make it more concise, focused, and engaging. Specifically, we have restructured the content to clearly distinguish the background, research gap, and objectives of the study, and we have removed discussion-like interpretations that were prematurely introduced. We believe the revised introduction now better aligns with academic conventions and improves the reader’s experience.

---

## [Decision Letter · Decision Letter 2]

Explainable Mortality Prediction Models Incorporating Social Health Determinants and Physical Frailty for Heart Failure Patients

PONE-D-23-36939R2

Dear Dr. Zhang,

We’re pleased to inform you that your manuscript has been judged scientifically suitable for publication and will be formally accepted for publication once it meets all outstanding technical requirements.

Kind regards,

Amir Hossein Behnoush

Academic Editor

PLOS ONE

Additional Editor Comments (optional):

Reviewers' comments:

Reviewer's Responses to Questions

**Comments to the Author**

Reviewer #7: All comments have been addressed

2. Is the manuscript technically sound, and do the data support the conclusions?

Reviewer #7: Yes

3. Has the statistical analysis been performed appropriately and rigorously?

Reviewer #7: Yes

4. Have the authors made all data underlying the findings in their manuscript fully available?

Reviewer #7: Yes

5. Is the manuscript presented in an intelligible fashion and written in standard English?

Reviewer #7: Yes

Reviewer #7: (No Response)

**Do you want your identity to be public for this peer review?** For information about this choice, including consent withdrawal, please see our Privacy Policy

Reviewer #7: No

---

## [Editor Report · Acceptance letter]

PONE-D-23-36939R2

PLOS ONE

Dear Dr. Zhang,

I'm pleased to inform you that your manuscript has been deemed suitable for publication in PLOS ONE. Congratulations! Your manuscript is now being handed over to our production team.

Kind regards,

on behalf of

Dr. Amir Hossein Behnoush

Academic Editor

PLOS ONE